# Applying Race and Ethnicity in Health Disparities Research

**DOI:** 10.3390/ijerph22101561

**Published:** 2025-10-14

**Authors:** Keith C. Norris, Matthew F. Hudson, M. Roy Wilson, Genevieve L. Wojcik, Elizabeth O. Ofili, Jerris R. Hedges

**Affiliations:** 1Division of General Internal Medicine and Health Services Research, David Geffen School of Medicine, University of California at Los Angeles, Los Angeles, CA 90024, USA; knorris@ucla.edu; 2Department of Medicine, School of Medicine—Greenville, University of South Carolina, Greenville, SC 29605, USA; matt.hudson@prismahealth.org; 3School of Medicine, Wayne State University, Detroit, MI 48202, USA; mrw@wayne.edu; 4Department of Epidemiology, Bloomberg School of Public Health, Johns Hopkins University, Baltimore, MD 21205, USA; gwojcik1@jhu.edu; 5Department of Medicine, Morehouse School of Medicine, Atlanta, GA 30310, USA; eofili@msm.edu; 6Department of Surgery, John A. Burns School of Medicine, University of Hawaii at Manoa, Honolulu, HI 98613, USA

**Keywords:** race, ethnicity, biomedical research, social and environmental drivers of health, health outcomes, health disparities, clinical trials, Research Centers for Minority Institutions (RCMI)

## Abstract

Health professionals commonly reference race and ethnicity to inform health care and administrative decisions. However, health researchers (and, arguably, society at large) misapply race and ethnicity when assuming an inherent relationship of these concepts with biological and health outcomes of interest. Misapplication of race potentially results from socially embedded identification predicated upon race essentialism, the belief that people within a racial group share “inherent” biological traits that define them as distinct from other racial groups. This false belief is often associated with implied racial hierarchies obscuring authentic causal disease relationships. Similarly, ethnicity is a socially and politically constructed group descriptor for people from a similar national or regional background who may share common cultural, historical, and social experiences. Thus, as for race, no inherent biological information is contained within such group definitions. This article summarizes the Research Centers for Minority Institutions (RCMI) 2025 Annual Grantee Meeting keynote session on Race and Ethnicity in Medicine. The session described how society originated and subsequently applied/misapplied race and ethnicity in various domains of policy and public health. The keynote session also considered the use of race and ethnicity in describing and envisioning biomedical research, clinical trials, clinical practice, and health services research. The authors summarize a more tenable use of race and ethnicity to advance biomedical research and health by focusing upon social and environmental drivers of health, population representation in clinical trials, and other factors. Associated recommendations from the keynote session are provided.

## 1. Introduction

The scientific community is increasingly scrutinizing the use of race and ethnicity as explanations for community health and clinical practice outcomes [1]. The fundamental definitions and use of race and ethnicity, particularly when explaining clinical practice outcomes, are receiving increasing attention. This summary of the Research Centers in Minority Institutions (RCMI) Annual Grantee Meeting keynote session on Race and Ethnicity in Medicine (Bethesda, MD, USA, March 2025) examined the historic and contemporaneous use and misuse of race and ethnicity in medicine. Advancing the 2024 National Academies of Sciences, Engineering, and Medicine (NASEM) report Rethinking Race and Ethnicity in Biomedical Research [1], we use this session report to clarify the concepts of race and ethnicity to encourage precise predictor identification and measurement essential to sound health science and clinical care.

Traditionally, basic science assumes a privileged role of seminal inquiry predicating essential knowledge in academic medicine. Investigators assume insights gleaned from basic science serve as predicates for biological phenomena such as disease manifestation, but health and disease are an intricate balance of many biological, social, and environmental factors. Consequently, it is reasonable to re-examine the definitions of race and ethnicity in health to both distinguish and appropriately integrate race and ethnicity as social constructs impacting biological processes and health. Basic science can benefit from this re-examination; clarification should redirect and refine a scientific focus on social and political factors impacting health and health care delivery rather than implying “race-based” genetic predispositions that protect or portend disease. Such an approach will encourage the biomedical sciences to incorporate more methodologically appropriate uses of social variables. A revised approach should lead to social and political science models that more precisely discern the interplay between underlying socioeconomic, political, and human biomedical disease factors and distributions across groups in the United States and elsewhere.

The National Institutes of Health (NIH) RCMI program espouses a core objective to “enhance the quality of all scientific inquiry and promote research on minority health and health disparities.” [2]. Unfortunately the quality of much human biomedical research from clinical trials to Artificial Intelligence informed real-world data analyses is limited by inconsistencies in how data on race and ethnicity are collected, as well as a lack of understanding of how these descriptors should be analyzed. We contend that fundamentally defining race and ethnicity is a prerequisite to accomplishing the RCMI objective of enhancing the quality of biomedical research for all. This narrative provides a brief overview of the historical context of race and ethnicity and provides a methodological grounding for the best use of race and ethnicity in public health/heath systems, medical research (including key considerations in the conduct of clinical trials) and clinical care delivery [3,4,5].

## 2. Materials and Methods

The 2025 RCMI Annual Grantee Meeting (Bethesda, MD, USA) included a keynote session on Race and Ethnicity in Medicine. The keynote session examined the historic and contemporaneous use and misuse of race and ethnicity in medicine. The authors served as keynote session panelists; they possess administrative and research leadership experience at minority serving institutions and/or service for the NASEM. Following a summary of the 2024 NASEM report, Rethinking Race and Ethnicity in Biomedical Research [1], panelists discussed implications of the report findings upon health disparities and related biomedical research.

This report summarizes the panel discussion and expands upon identified issues, including the historical genesis of race and ethnicity and the use of race and ethnicity in health services research. The authors summarize a more tenable use of race and ethnicity to advance biomedical research and health by focusing upon social and environmental drivers of health, representation in clinical trials, and other factors. The Section 4 also provides the panelists’ associated recommendations for the biomedical application of race and ethnicity concepts.

## 3. Results

### 3.1. Defining Race and Ethnicity

Race and ethnicity define and distinguish subgroups of societal members. However, these groupings have no biological basis, nor do these groups biologically stratify humans into non-existent subspecies. The more accurate terminology of racial and ethnic identities reinforces the true social basis of these terms. Per LaVeist, race in America is an inconsistent, socially assigned latent, proxy construct that conveys no intrinsic biological differences except for select physical attributes deemed contextually relevant [4].

Ethnicity categorizes people based on shared social factors or cultural traits (including language, religion, customs, and shared history) [6]. Like race, scientists have used ethnicity as a proxy for social and environmental factors. Neither race nor ethnicity are inherent predictors of disease or therapy response [1,3,4,5,7]. Rather, race and ethnicity function as proxies (albeit poorly) for the social and environmental drivers of health, and other forces that drive maldistribution of wellness and disease among socially identified groups. These distinctions are further blurred when race labels encompass multiple ethnicities. An example of this would be to speak of Pacific Islanders, which encompasses not only groups with Polynesian ancestry, but also multiple other groups having Micronesian ancestries with their own distinct languages and cultural practices. Similarly, although Judaism is a religion, those with Jewish ancestry (whether practicing Judaism or not) have at times been grouped socio-politically as an ethnic group. Nazi Germans even advanced the false narrative that Jewish ancestry represented a racial group.

These race-correlative factors may override or trigger an individual’s biological disposition. For instance, race historically predicts housing options and neighborhood resource constitution [8,9]. These social conditions more precisely predict health than race. In fact, White persons residing in predominately Black communities often suffer from the community’s suboptimal resource allocation [10]. Thus, race may lack a “protective effect” when select social conditions drive neighborhood health status, health care access and treatment outcomes [11,12,13].

### 3.2. Historical Context and Evolution of Race and Ethnicity in Society and Science

The concepts of race and ethnicity as categories for groups of humans have evolved. Centuries ago, society distinguished groups of people by language, nationality, culture, and other factors. In the 18th century, European scientists, most notably Carl Linnaeus, the father of Modern Taxonomy, formalized a hierarchical classification of groups of people based on physical appearance and continent of origin [14]. Linnaeus also added presumed innate personal attributes to each group, creating not only the concept of race but racism.

The spread of European colonialism, including the establishment of the United States, was associated with the use of religious and science-based applications of race to define human subspecies and justify resource acquisition [15]. This misguided belief became embedded within the broader European and colonial cultures, justifying racialized laws, policies, and practices in many countries [6]. In the United States, Germany, and other nations, these practices ranged from slavery to eugenics and genocide [6]. The dominant culture predicated the oppression of other races on the belief that certain human subspecies were “organically” ordained, per “scientifically proven” racial hierarchies [6].

Ethnicity is a socially and politically constructed term for a population group that describes people from a similar national or regional background. Individuals in such groups often also share common cultural, historical, and social experiences. Individuals often identify with or are assigned to an ethnic group based on a belief in shared ideas, values, behaviors, heritage, or language [1,16]. Like race, ethnicity has no inherent biological traits other than superficial inconsistent phenotypical appearances. Given the common confusion of ethnicity with race and the analogous misapplication of ethnicity as a biological construct, much of our discussion regarding race misuse is applicable to misuse of ethnicity in biomedical research.

The Human Genome Project confirmed that there are no distinct racial or ethnic group differences within the *Homo sapiens* species, and that any two individuals from within a historically defined racial group are likely to have more genetic variation than two individuals selected across groups [17]. Despite an abundance of scientific evidence confirming the lack of any scientific foundation for innately discrete groups of people, as well as consensus that race and ethnicity as commonly used are defined by social identity, there remains a deep-seated resistance to recognizing race and ethnicity as social constructs. Not only does the belief that people within a racial group share “inherent” biological traits that define them as distinct from other racial groups (race essentialism) persist in the biomedical sciences and health professions, but some extend this false belief to generate biological hierarchal biases [18]. Other aspects of personal identity—sex, sexual identity, religion, ableness, and income status—also have been predicates for discrimination and structural barriers; biases toward such personal identity factors accompany racism in discrimination potency [6,18].

### 3.3. Racism

Racism is a system that allocates resources and opportunities based on racial designation, granting advantages to some groups and disadvantages other groups, but on the whole saps the strength and diminishes the potential of a society [19,20]. Exacerbating the challenge in identifying and confronting racism are beliefs related to the intersection of race with other individual identities and characteristics including differences in physical appearance such as skin color. For example, skin color has been viewed by some in society as synonymous with race. Variation in skin color does not follow a clear distribution based on racial and ethnic categories, but in the United States and many other countries, skin color has been reported to modify the level of perceived discrimination, not only across but within racial or ethnic groups. As noted above, intersectionality between race and a myriad of individual identities may be linked to varying degrees of stigma, bias, and discrimination independent of race or in addition to race, such as immigration status, sexual orientation, religion, nationality, and more. Racism along with other identity-based forms of discrimination and prejudice may be linked to policies and practices which more directly explain health or illness in population groups. Racism begets discrimination and prejudice which more directly explain health or illness in racially defined groups. Access to poor health care, education, employment, and food quality, along with exposure to environmental toxins and more, may act as independent root causes for selected conditions and in totality often trigger maladaptive neurocognitive and biological pathways and epigenetic changes adversely impacting health and health outcomes [21]. Consequently, we encourage scientists to unravel relationships between race, racism, ethnicity, intersectional factors, social and environmental drivers of health, and biological disposition.

### 3.4. Race and Biology

Genotype and geo-evolutionary forces interact to produce human variation, a small subset of which are commonly labeled as phenotypes determining different races, dependent on social context [17]. However, we reiterate that no innate biological differences exist between race and ethnicity defined groups. Group-level differences may exist in the prevalence of select genetic variants predisposing individuals to cystic fibrosis, breast cancer mutations, and sickle cell anemia. However, none of these variants defines, or is exclusive to, any racial or ethnic group. While scientists can trace some of these gene variants to a “founder effect” in small populations or to environmental exposures (e.g., infectious diseases, intense ultraviolet light exposure), these rarely align with broad racial categories. Despite this, these group-level differences have contributed not only to the conflation of race and biology [17], but the broad racialization of genetics [22].

Scientists have long observed group-level racial and ethnic differences in disease and biomarker distributions [23,24]. However, the observed differences do not arise from innate biological differences between these broad groupings of humans. Rather, scientists can explain many differences through social and environmental drivers of health (e.g., limited access to health care or wellness resources). Further, psychological stress potentially portends differential activation of neurocognitive systems and pathways including maladaptive health behaviors (e.g., eating disorders, substance use, etc.), biological systems, and pathways including activation/suppression of neurohormonal, oxidative stress, inflammatory and immune systems, and epigenetic changes, ultimately manifesting as undesired health states (Figure 1) [21]. Therefore, biology as well as socioeconomic and stress factors linked to racial groupings and disease may interact at multiple levels.

### 3.5. Racial and Ethnic Identity and Genetic Ancestry as Data

Methodologists typically regard race and ethnicity as nominal or categorical variables. Racial categories are correlated with social and political factors more strongly than correlated with health conditions of interest (e.g., rates of hypertension, diabetes, human immunodeficiency virus infection, and cancer). Race, as a discrete social construct, holds no inherent biological attributes directly related to any medical or physiological condition like blood pressure or hemoglobin A1c levels [1,4]. Consistent with the movement away from “race-based medicine,” we must note that race categories may capture a group of people who experience select clinical circumstances but does not explain disease or serve as a guide to direct therapy [1,4].

When trying to capture potential inherent biological differences, the use of genetic variation instruments, such as “ancestry-informative markers” (AIMs), have been proposed. Such instruments estimate the proportion of a specific individual’s DNA variation that aligns with a reference dataset curated to represent geographic origin and “genetic ancestry.” [17]. While genetic instruments more precisely represent the ancestral origins of an individual than race and ethnicity, multiple issues inhibit their value as a proxy for race and ethnicity. First, continental or large-area genetic ancestry does not equal race or ethnicity. While some overlap exists, genetic similarity and race and ethnicity remain distinct constructs capturing different underlying forces. Most importantly, while these genetic instruments of ancestry may have associations with health conditions and gene variants, they have no inherent biological information as they are selected agnostically across the genome [17]. In fact, genetic ancestry associations with health often reflect the substructure of environmental exposures, history, anthropology, politics, sociology, etc., that evolves over many years [17]. Thus, for health purposes, estimated genetic ancestry, just like race and ethnicity, lack utility for dictating a specific therapeutic approach or serving as an individual-level modifier in clinical formulas or algorithms.

### 3.6. Race and Ethnicity with Health Disparities

Health disparities are largely preventable health differences that adversely affect populations who experience greater challenges to optimal health. These health disparities are closely linked with intergenerational social, economic, and/or environmental disadvantages [24]. Typically, health disparities in the United States have historically been associated with racial and ethnic groups impacted by lower social resource access and in lower socioeconomic status communities. Disparities may also be found with other population groupings using other categories such as sex, religion, sexual preferences, and ableness. These disparities may be driven by group-level inequities in access to health-affirming (social, economic, environmental, and legal) resources and opportunities [24].

Race and ethnicity-based health disparities research typically studies the groups with the greatest proportional burden of a given disease and presume disparity-mitigating interventions will benefit all groups. Importantly, interventional studies and clinical trials conducted with a diverse patient sample can clarify whether an intervention or treatment has non-biological comparative effectiveness among groups and may suggest the need for tailoring the approach for different groups or contexts [25,26].

The Consolidated Framework for Implementation Research (CFIR) [27,28] is a structured approach to interventional research that heightens attention to the context driving the subsequent uptake of effective health care interventions. Such frameworks may more readily explain disease distribution and intervention impact among racial and ethnic groups, thereby helping to address many health disparities. For example, the “Inner Setting” concept embeds the importance of culture-shared values, beliefs and norms across the health care system or delivery points. Such values include human equality-centeredness (the inherent equal worth and value of all human things). Similarly, CFIR considers intervention impact as a function of individual characteristics such as capability, opportunity, and motivation. These “individual” considerations are potentially salient both for the care systems serving a racially diverse population and the patient populations themselves. From an intervention perspective, factors such as literacy (health related and otherwise), household income, number and status of dependents, migration status, level of employment, level of education, and housing status as well as health-related social needs more explicitly explain health disparities than race and ethnicity. Indeed, race and ethnicity may function as an imprecise proxy for many social and environmental drivers of health. To this end the RCMI community emphasizes multi-level inquiry—examining social and environmental drivers of health at the individual, community, and societal levels—to bridge the gap between health disparities documentation and the public health and health system intervention [2,29].

### 3.7. Race and Ethnicity in Clinical Research

Clinical researchers make efforts to document sociodemographic data (e.g., sex, geographic location, race and ethnicity) to examine sample representativeness, a desired feature advancing the study’s external validity. Some scientists conflate racial and ethnic sample representativeness with genetic sample representativeness to assess the capture of genetic variation. This is not advised given the high levels of genetic substructure within these social constructs, as well as the exclusion of some groups with commonly used racial and ethnic categories. Additionally, and most importantly, race and ethnicity are not predictors of the presence or absence of clinically relevant genetic variation. Unfortunately, such an approach only serves to motivate and perpetuate medical racialization and racial/ethnic stereotyping [22]. By contrast, ensuring racially and ethnically representative study participants may introduce more variation in underlying social and environmental influences. Such influencing factors may introduce variations in activation of neurocognitive and biological pathways/systems in the cohort (as modeled in Figure 1). These phenomena may be more powerful than genetic differences in ultimately predicting disease expression and intervention impact. Thus, we should move to collect data on social exposures and environmental conditions which more explicitly and accurately explain health status and intervention impact than race and ethnicity which are merely individual-level proxies.

## 4. Discussion

The RCMI is dedicated to reducing all forms of health disparities while improving health outcomes for all. To this end, many RCMI projects target the social and environmental drivers of health. Project investigators often undertake efforts to reduce such factors influencing health in partnership with patients and community stakeholders who are particularly positioned to inform and refine approaches to reduce such non-medical factors impacting health. Policy-related interventions are major priorities, as policy may both enhance intergenerational well-being and positively impact broader societal goals. Policy-centric interventions may ensure a healthier nation thereby enhancing national and economic security [30].

The RCMI approach highlights addressing health-related social needs (HRSN) [31]. This comprehensive interventional care model describes (a) the social and environmental upstream factors, (b) the midstream patient support factors (individual social needs), and (c) downstream patient care factors (which include medical, behavioral, and other health system interventions). Downstream care needs are a function of upstream factors. Various tactics to address health disparities must include health policy (upstream), social services screening and support (midstream) and individual health interventions (downstream) [31].

### Recommendations for Future Health Disparities Research

Fundamentally recognize and appropriately define race as a social construct. This approach divorces scientists from studying race as an immutable characteristic begetting invasive clinical intervention to change/modify genetic predisposition. As race and ethnicity operate as proxy variables, their value as part of a sociodemographic profile is best suited for population level assessments in which their utility as a proxy is well-justified. Where possible, the focus of the research question and data collection should be upon the presumed social and environmental drivers of health that are more causally associated with a given disease than race or ethnicity. This approach is consistent with the U.S. Office of Management and Budget (OMB)-published revisions to Statistical Policy Directive No. 15: Standards for Maintaining, Collecting, and Presenting Federal Data on Race and Ethnicity (SPD 15) [32]. The categories in the OMB standards are understood to be socio-political constructs, not an attempt to define race and ethnicity biologically or genetically. The standards have seven minimum categories for data on race and ethnicity: American Indian or Alaska Native, Asian, Black or African American, Hispanic or Latino, Middle Eastern or North African, Native Hawaiian or Pacific Islander, and White.

Tailor the use of race and ethnicity descriptors to the specific study design and goals. For a given study, researchers should define race and ethnicity, and how and why they are conceptualized and measured, given the research hypothesis. When possible, researchers should disaggregate racial and ethnic groups to better capture within-group variations (e.g., Black Americans stratified by low, middle, and high income, or nationality within racial or ethnic groups). Studies planning for new data collection should also endeavor to collect more granular data allowing for multi-racial and multiethnic participant assessments. Researchers should clarify the analytic plan and data reporting to encourage data consistency and allow cross-study comparisons. Finally, each study report should articulate the limitations of racial and ethnic data analysis in the study. Researchers should explicitly identify data limitations and temper inferences accordingly, especially when deliberating insights gleaned from legacy datasets [3].

Incorporate community involvement in research study design and implementation. Racial and ethnic discrimination and biases in medicine have been lessened through innovative community and local policy approaches to address the social and environmental drivers of health and HRSN and help alleviate harm [33]. To this end researchers should consider engaging communities in study design, data collection, results interpretation, and evidence dissemination [1]. Here, we define community as “a commonality of desire, a communion with those around us, an extension of the bonds of kin and friend to all those who share a common fate with us.” [34]. This definition underscores investigators’ and community members’ shared responsibility and their increased value in shared problem prioritization, problem address, results interpretation, and information dissemination.

Community-engagement approaches can build trust, ensure transparency, and in many instances beget sustained interdisciplinary partnerships incorporating data sovereignty [1,3,35,36]. A community-engaged approach can also improve communication, promote respect, and optimize essential operational strategies (e.g., time required for successful outreach, clarifying the rationale for participant refusal, approach to revise consent forms, correct recruitment materials, and identification of appropriate/accessible study implementation site) [37,38,39]. The latter is particularly relevant as many communities lack access to quality health care facilities and, by extension, to clinical research sites [40].

Strive for extensive racial and ethnic representation in health studies. Academic interventions to reduce health disparities are best conducted with broad participant representation to both improve the scientific quality by capturing a wider range of socially and environmentally induced biological consequences across populations, as well as to increase confidence among participating groups that the research findings will benefit them and their communities. RCMI sites have performed this through the use of a community-engagement core which fosters involvement in health project design, implementation, data analysis, and reporting [2].

Avoid racial and ethnic categorization for clinical decision-making. At a clinical level, take caution when considering racial or ethnic group-level associations in disease prevalence or clinical outcomes or group stereotypes, as such associations may fail to explain outcomes of individual-level clinical care [9]. Categories are best used to complement a patient’s social history and lived experience alongside social networks, social and environmental drivers of health, and family history (which may elucidate hereditary risk). As such, racial or ethnic identity may more broadly help inform care plans (e.g., follow-up visit frequency, structurally and culturally tailored lifestyle recommendations, social support referrals, and other relevant HRSN interventions). Importantly, investigators should not use race or ethnicity to exclude or to make a diagnosis, nor to inform a specific drug therapy or as modifiers in individual-level formulas or algorithms [9].

## 5. Conclusions

As directed by newer NIH priorities, health disparities research should address root causes of disease. To address population health disparities, it is essential to ensure that the distribution of life- and health-affirming resources and opportunities are accessible fairly across the population avoiding racially, ethnically or other hierarchical group priorities. Prior to the civil rights act in 1964, federal laws, policies, and practices were often specifically discriminatory by population group. Currently, alternative discrimination by proxy (e.g., neighborhood, socioeconomic status) may continue to disadvantage marginalized racial and ethnic groups [30].

Of note, this article has not addressed a multitude of historical factors which have been associated with discriminatory practices and ongoing socioeconomic differences between population groups in the United States. We also did not include sociopolitical models which have been developed to explain these historical factors. While continued discrimination and socioeconomic differences must be addressed in the sociopolitical arena, this article focused on addressing health disparities through the correct application of race and ethnicity within human biomedical research.

Our discussion proposes a more tenable approach to race and ethnicity definitions relative to addressing health services and health disparities. This approach can advance biomedical research and health by directing focus to more salient factors such as social and environmental drivers of health and population representation in clinical trials. Based upon the above review of race and ethnicity in biomedical research and health care delivery, the authors provide the following recommendations for guarding against misuse of labels, such as race and ethnicity, in biomedical research and health care:Fundamentally recognize and appropriately define race as a social construct.Tailor the use of race and ethnicity descriptors to the specific study design and goals.Incorporate community involvement in research study design and implementation.Strive for extensive racial and ethnic representation in health studies.Avoid racial and ethnic categorization for clinical decision-making.

## Figures and Tables

**Figure 1 ijerph-22-01561-f001:**
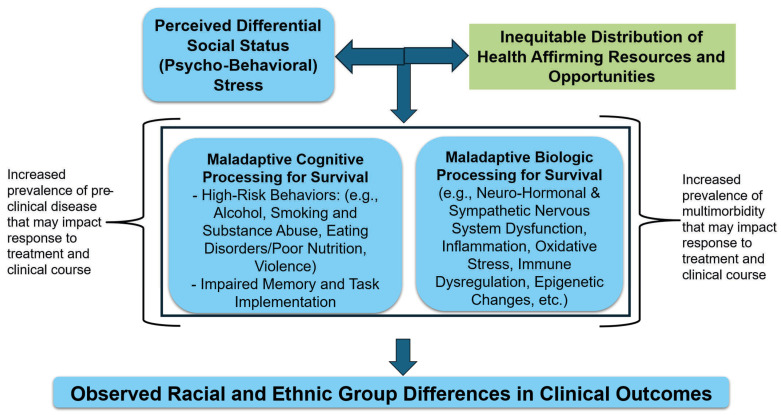
Mechanistic Drivers of Health. The impact of psycho-behavioral stress can be compounded by insufficient health-affirming resources and opportunities. Chronic psycho-behavioral stress may trigger the maladaptive behavioral and neuro-inflammatory drivers of health disparities.

## Data Availability

No new data were generated for this report other than as summarized in the article.

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
