# Peer review of "Applying Race and Ethnicity in Health Disparities Research"

_ijerph, 2025, doi:10.3390/ijerph22101561_

Round 1
Reviewer 1 Report
Comments and Suggestions for Authors
This paper summarizes a keynote session of a scientific meeting on race and ethnicity in medicine and public health.
Major comment: I note that the article does not mention the new race/ethnicity standards in OMB Statistical Policy Directive No. 15 (SPD15) on the use of a combined race/ethnicity question, new Middle East and North Africa reporting category and requirement to collect detailed race/ethnicity responses. Was this not discussed as part of this session? Even if it was not discussed, it seems to be something that the authors of this paper need to comment on since it will likely have a major impact on the topic at hand.
Minor suggestions:
Line 19 - the subject of this sentence, "health care authors", could benefit from a better term.
Line 22 - the term "race essentialism" is used here and elsewhere without further definition or explanation. It is an important concept and deserves to be defined and briefly explained for the benefit of readers who may be as familiar with this topic.
Line 38-9 - This sentence needs at least one reference.
Line 60 - It would useful to provide more details on the "social and political science models" referred to here. What are some examples? References here would be helpful, I think.
Line 305 - this sentence seems to be incomplete as it ends with the adjective "interdisciplinary" without further explanation regarding what this is referring to.
Author Response
Reviewer #1: This paper summarizes a keynote session of a scientific meeting on race and ethnicity in medicine and public health.
Major comment: I note that the article does not mention the new race/ethnicity standards in OMB Statistical Policy Directive No. 15 (SPD15) on the use of a combined race/ethnicity question, new Middle East and North Africa reporting category and requirement to collect detailed race/ethnicity responses. Was this not discussed as part of this session? Even if it was not discussed, it seems to be something that the authors of this paper need to comment on since it will likely have a major impact on the topic at hand.
We appreciate the recommendation of the reviewer. While not the focus of the panel discussion, we have added information related to the OMB standards to our first discussion recommendation as highlighted below:
Fundamentally recognize and appropriately define race as a social construct. This approach divorces scientists from studying race as an immutable characteristic begetting invasive clinical intervention to change/modify genetic predisposition. As race and ethnicity operate as proxy variables, their value as part of a sociodemographic profile is best suited for population level assessments in which their utility as a proxy is well-justified. Where possible, the focus of the research question and data collection should be upon the presumed social and environmental drivers of health that are more causally associated with a given disease than race or ethnicity. “This approach is consistent with the U.S. Office of Management and Budget (OMB) published revisions to Statistical Policy Directive No. 15: Standards for Maintaining, Collecting, and Presenting Federal Data on Race and Ethnicity (SPD 15), [32] The categories in the OMB standards are understood to be sociopolitical constructs, not an attempt to define race and ethnicity biologically or genetically. The standards have seven minimum categories for data on race and ethnicity: American Indian or Alaska Native, Asian, Black or African American, Hispanic or Latino, Middle Eastern or North African, Native Hawaiian or Pacific Islander, and White.”
Minor suggestions:
Line 19 - the subject of this sentence, "health care authors," could benefit from a better term.
We have changed the term to “health researchers.”
Line 22 - the term "race essentialism" is used here and elsewhere without further definition or explanation.
The abstract has been modified to include a definition for race essentialism as shown below. We also provided this definition where race essentialism is introduced in the body of the article.
“Misapplication of race potentially results from socially embedded identification predicated upon race essentialism, the belief that people within a racial group share “inherent” biological traits that define them as distinct from other racial groups. This belief is often associated with implied racial hierarchies obscuring authentic causal disease relationships.”
Line 38-9 - This sentence needs at least one reference.
Although we introduce the National Academies of Science, Engineering & Medicine (NASEM) report on Race and Ethnicity later in this paragraph, we will also introduce that citation here as the NASEM report is relevant to these statements.
Line 60 - It would be useful to provide more details on the "social and political science models" referred to here. What are some examples? References here would be helpful, I think.
As shown below, we have modified the statement to more accurately state our intent to stimulate more precise disease modeling within populations, not to discuss those models in this article. Although such modeling has begun (as noted by bracketed comments below the modified sentence), the purpose of our article was not to describe or critique such models.
“A revised approach should lead to social and political science models that more precisely discern disease causes and distributions across groups in the United States and elsewhere.”
[Note that Griffith et al. have described a theoretic framework behind dismantling racism to reduce health disparities (Griffith DM, Mason M, Yonas M, Eng E, Jeffries V, Plihcik S, et al. Dismantling institutional racism: theory and action. American Journal of Community Psychology. 2007;39(3-4):381-92.); while Myers (Myers HF. Ethnicity- and socio-economic status-related stresses in context: an integrative review and conceptual model. Journal of behavioral medicine. 2009;32(1):9-19) proposed a conceptual model that takes into consideration the interaction of ethnicity, socioeconomic status, and psychosocial adversities, as well as ethnicity-related stresses that lead to biopsychosocial vulnerabilities. Hill and colleagues, developed an interdisciplinary biopsychosocial model that led to a framework for investigators to consider the level of disparities-focused analyses (Hill CV, Perez-Stable EJ, Anderson NA, Bernard MA. The National Institute on Aging Health Disparities Research Framework. Ethn Dis. 2015;25(3):245-54, Crimmins EM, Seeman TE. Integrating biology into the study of health disparities. Population and Development Review. 2004;30(1):89-107). Moreover, Bruce and colleagues extended and integrated work by Myers (Myers HF. Ethnicity- and socio-economic status-related stresses in context: an integrative review and conceptual model. Journal of behavioral medicine. 2009;32(1):9-19.) and Seeman and Crimmins (Crimmins EM, Seeman TE. Integrating biology into the study of health disparities. Population and Development Review. 2004;30(1):89-107.) to describe how sociological factors such as racism create stressful social environments that ultimately shape behavioral and physiological responses thereby adversely impacting not only health and well-being, but even mortality (Bruce MA, Griffith DM, Thorpe RJ, Jr. Stress and the kidney. Advances in chronic kidney disease. 2015;22(1):46-53).]
Line 305 - this sentence seems to be incomplete as it ends with the adjective "interdisciplinary" without further explanation regarding what this is referring to.
Thank you. A portion of the sentence was inadvertently deleted during final manuscript preparation. The sentence now reads as follows:
“Community-engagement approaches can build trust, ensure transparency, and in many instances beget sustained interdisciplinary partnerships incorporating data sovereignty.”
Reviewer 2 Report
Comments and Suggestions for Authors
Major concerns
- There is no information on ethnicity as if it is the same concept of race.
- It is unclear about what the issues with quality of biomedical research now and how future research should be improved.
- The suggestions such as considering upstream factors are not innovative.
Minor concerns
- Please be more specific about types of biomedical research that you are suggesting.
- Please talk about issues like skin color vs. racial discrimination.
- Please talk about issues like migration and sexual orientation that are intertwined with stereotype, stigma, bias, and discrimination.
- Please provide more details on white supremacy culture and how future research should provide evidence on it.
Author Response
Reviewer #2:
Major concerns:
There is no information on ethnicity as if it is the same concept of race.
We have modified the abstract to address this point as follows and have similarly provided a more expanded discussion related to ethnicity in section 3.2.
“Similarly, ethnicity is a socially and politically constructed group descriptor for people from a similar national or regional background who may share common cultural, historical, and social experiences. Thus, as for race, no inherent biological information is contained within such group definitions.”
It is unclear about what the issues with quality of biomedical research now and how future research should be improved.
We now address this more explicitly in the third paragraph of the introduction as follows:
“The National Institutes of Health (NIH) RCMI program espouses a core objective to “enhance the quality of all scientific inquiry and promote research on minority health and health disparities.” [2] Unfortunately the quality of much biomedical research from clinical trials to Artificial Intelligence informed real-world data analyses, is limited by inconsistencies in how data on race and ethnicity are collected, as well as a lack of understanding of how they should be analyzed. We contend that fundamentally defining race and ethnicity is a prerequisite to accomplishing the RCMI objective of enhancing the quality of biomedical research for all. This narrative provides a brief overview of the historical context of race and ethnicity and provides a methodological grounding for the best use of race and ethnicity in public health/heath systems, medical research (including key considerations in the conduct of clinical trials) and clinical care delivery. [3-5]”
The suggestions such as considering upstream factors are not innovative.
While we agree that considering upstream factors are not innovative for many people working in health disparities and minority health, there remains a larger audience that are still unaware of mechanisms creating associations between health outcomes and social determinants of health and related factors. Thus, we believe it is important to include such information.
Minor concerns:
Please be more specific about types of biomedical research that you are suggesting.
Please see the above addition to the introduction as noted above.
Please talk about issues like skin color vs. racial discrimination. PLUS
Please talk about issues like migration and sexual orientation that are intertwined with stereotype, stigma, bias, and discrimination.
We have added the following expansion under section 3.3 Racism.
“Exacerbating the challenge in identifying and confronting racism are beliefs related to the intersection of race with other individual identities and characteristics including differences in physical appearance such as skin color. For example, skin color has been viewed by some in society as synonymous with race. Variation in skin color does not follow a clear distribution based on racial and ethnic categories, but in the United States and many other countries, skin color has been reported to modify the level of perceived dis-crimination, not only across but within racial or ethnic groups. As noted above, intersectionality between race and a myriad of individual identities may be linked to varying degrees of stigma, bias, and discrimination independent of race or in addition to race, such as immigration status, sexual orientation, religion, nationality, and more. Racism along with other identity-based forms of discrimination and prejudice may be linked to policies and practices which more directly explain health or illness in population groups.”
Please provide more details on white supremacy culture and how future research should provide evidence on it.
The social theory of White supremacy culture is beyond the context of this article, but as suggested by the reviewer, has been used to characterize different access to health affirming care for majority versus minority cultures in countries with a predominately White population. The basis for racism in the United States as outlined earlier in the article, including documented discriminatory medical practices, appear to have been driven by a false presumption of the superiority of Whites over other groups. Some sociologists are concerned that majority cultures will unwittingly continue discriminatory practices. We have added the following statement to the conclusion.
“As directed by newer NIH priorities, health disparities research should address root causes of disease. To address population health disparities, it is essential to ensure that the distribution of life- and health-affirming resources and opportunities are accessible fairly across the population avoiding racially, ethnically or other hierarchical group priorities. Prior to the civil rights act in 1964, federal laws, policies and practices were often specifically discriminatory by population group. Currently, alternative discrimination by proxy (e.g., neighborhood, socioeconomic status) may continue to disadvantage marginalized racial and ethnic groups.[41] Correct application of race and ethnicity in biomedical research can help us address such challenges.”
Round 2
Reviewer 2 Report
Comments and Suggestions for Authors
- Line 67: Please elaborate what social and political science models you try to provide indications to.
- Line 72: The definition of biomedical research is unclear. For example, animal research about a drug for cattle does clinical trials but does not require race and ethnicity.
- In the 3.1 section, could you provide specific examples such as a particular racial group might include multiple ethnic populations?
- Also, Jew is a religion in certain areas but a race in other areas. This kind of specific example should be discussed and explained clearly to make people better understand the difference.
- In the 3.4 section, could you link this issue to white supremacy?
- In the 3.6 section, could you elaborate issues like model minority stereotypes?
- The authors need to better articulate results and discussions. It does not feel like enough information on the association between the history of race and ethnicity and the current socioeconomic status.
- Line 578-583: Please provide a successful example of how to increase the representation of different racial and ethnic groups.
- Discussions: I feel most advices have been heard too many times. It will be helpful if the authors could provide some successful examples to make the audience learn from it.
Author Response
We are resubmitting this Perspectives article for the IJERPH which reports on a keynote session at the 2025 RCMI Investigators meeting which included a keynote presentation and panel discussion by the authors.
We are most appreciative of the comments and suggestions from the two reviewers regarding our submission entitled “Applying Race & Ethnicity in Health Disparities Research.” We have made changes to the article in response to this input. We address those changes related to the reviewers’ comments and suggestions in more detail below. Again, we show the changes as underlined text.
Please note that we have not been successful in locating a better-quality Figure 2. Therefore, we have summarized the content of Figure 2 in the text and deleted that figure and figure legend.
Second-Round Reviewer Comments – Reviewer #2
1. Line 67: Please elaborate what social and political science models you try to provide indications to.
This statement beginning with line 67 has been revised as follows:
“A revised approach should lead to social and political science models that more precisely discern the interplay between underlying socioeconomic, political, and biomedical disease factors and distributions across groups in the United States and elsewhere.”
The intent of the statement (line 67) is not to advance a specific (existing or future) social or political science model for explaining observed health disparities between population groups, but rather to clarify that when developing or applying such models, assigning intrinsic biological traits to race or ethnicity obscures the relative roles of presumed underlying causative factors. Factors considered in social and political models are likely to also have regional and temporal elements that also will need to be teased out. Such discussion is beyond the goals of this article.
- Line 72: The definition of biomedical research is unclear. For example, animal research about a drug for cattle does clinical trials but does not require race and ethnicity.
The reviewer correctly identifies an area warranting further clarity. We have edited the sentence beginning on line 72 as follows:
“Unfortunately, the quality of much human biomedical research from clinical trials to Artificial Intelligence informed real-world data analyses is limited by inconsistencies in how data on race and ethnicity are collected, as well as a lack of understanding of how they should be analyzed.”
- In the 3.1 section, could you provide specific examples such as a particular racial group might include multiple ethnic populations?
We have inserted the following information in this section:
“These distinctions are further blurred when race labels encompass multiple ethnicities. An example of this would be to speak of Pacific Islanders which encompasses not only groups with Polynesian ancestry across the Pacific, but also multiple other groups having Micronesian ancestries with their own distinct languages and cultural practices.”
- Also, Jew is a religion in certain areas but a race in other areas. This kind of specific example should be discussed and explained clearly to make people better understand the difference.
We have inserted the following information in this section:
“Similarly, although Judaism is a religion, those with Jewish ancestry (whether practicing Judaism or not) have at times been grouped socio-politically as an ethnic group. Nazi Germans even advanced the false narrative that those with Jewish ancestry represented a racial group.”
- In the 3.4 section, could you link this issue to white supremacy?
Section 3.4 reinforces the distinction between the concept of “race” and biology. The term “white supremacy” refers to a racist ideology that asserts the superiority of white people and seeks to maintain that status. The reviewer may also have been referring to “white supremacy culture” theory which describes widespread and often invisible societal norms presumed to elevate “whiteness.” This ideology and the accompanying theory of hidden social “norms” perpetuating racism have become popular themes in the study of racism. While, understanding how societal factors may contribute to racism is important work, the intent of this article is focused on debunking the assumptions which have falsely tied “race” as a biological factor to the study of human health disparities.
- In the 3.6 section, could you elaborate issues like model minority stereotypes?
Section 3.6 highlights the multifactorial basis for health disparities across US population subgroups. Per Wikipedia, “the term model minority refers to a minority group, defined by factors such as ethnicity, race, or religion, whose members are perceived to be achieving a higher socioeconomic status in comparison to the overall population average. Consequently, these groups are often regarded as a role model or reference group for comparison to external groups (outgroups)… The concept of the model minority has generated controversy due to its historical application to suggest that economic intervention by governments is unnecessary to address socioeconomic disparities among particular racial groups.”
The rationale for introducing the “model minority stereotype” to section 3.6 is unclear. It is not our intent to imply that resources which influence the social determinants of health should be withheld from any population subgroup; quite the opposite is true. We believe that our statements are contrary to any support for the concept of a “model minority stereotype.” Further, the introduction of that concept, especially given its political attribution, in this article focusing on health disparities is likely to be confusing to the biomedical researcher.
- The authors need to better articulate results and discussions. It does not feel like enough information on the association between the history of race and ethnicity and the current socioeconomic status.
As noted in points #5 & 6 above, the reviewer has identified some significant sociopolitical models and theories which many social and political scientists believe have contributed to discrimination in the United States. Similarly, we believe that when such discrimination contributes to socioeconomic group differences in the United States, these can in turn contribute to observed health disparities. Some of these factors influencing socioeconomic states we mention and also provide citations for more in-depth reading. The reviewer advocates for a more extensive discussion of racism and sociopolitical models which have been used to explain the nature of discrimination and resulting socioeconomic differences. In our attempt to keep the paper concise and focused on actionable items which can be used to address health disparities, we have not expanded the paper to include a broader discussion of racism and its origins. To acknowledge that there are additional elements which we did not develop to the fullest extent possible, we have added the following paragraph to the conclusion section.
“Of note, this article has not addressed a multitude of historical factors which have been associated with discriminatory practices and ongoing socioeconomic differences between population groups in the United States. We also did not include sociopolitical models which have been developed to explain these historical factors. While continued discrimination and socioeconomic differences must be addressed in the sociopolitical arena, this article focused on addressing health disparities through the correct application of race and ethnicity within human biomedical research.”
- Line 578-583: Please provide a successful example of how to increase the representation of different racial and ethnic groups.
In the discussion where we advocate for representativeness in clinical studies, we have included a statement discussing the role of the RCMI community engagement core (present at each RCMI site) as an example, as follows:
“Strive for extensive racial and ethnic representation in health studies. Academic interventions to reduce health disparities are best conducted with broad participant representation to both improve the scientific quality by capturing a wider range of socially and environmentally induced biological consequences across populations, as well as to increase confidence among participating groups that the research findings will benefit them and their communities. RCMI sites have done this through the use of a community-engagement core which fosters involvement in health project design, implementation, data analysis, and reporting. [2]”
- Discussions: I feel most advices have been heard too many times. It will be helpful if the authors could provide some successful examples to make the audience learn from it.
Our intent has been to represent the conference keynote and panel presentations and therefore keep the focus of the article on addressing health disparities through the correct application of race and ethnicity within human biomedical research. We reviewed our specific recommendations in Section 4.1and each recommendation is accompanied by practical guidance and an accompanying reference(s). We acknowledge that other examples could be provided in the future and might form the substance of a follow-up methodological paper.